# Running Propensities of Athletes with Hamstring Injuries

**DOI:** 10.3390/sports7090210

**Published:** 2019-09-12

**Authors:** Dai Sugimoto, Brian D. Kelly, David L. Mandel, Duncan A. d’Hemecourt, Sara C. Carpenito, Charles A. d’Hemecourt, Pierre A. d’Hemecourt

**Affiliations:** 1Division of Sports Medicine, Department of Orthopedics, Boston Children’s Hospital, Boston, MA 02115, USA; David.Mandel@childrens.harvard.edu (D.L.M.); Duncan.dHemecourt@childrens.harvard.edu (D.A.d.); Charles.dHemecourt@childrens.harvard.edu (C.A.d.);; 2The Micheli Center for Sports Injury Prevention, Waltham, MA 02453, USA; 3Harvard Medical School, Boston, MA 02115, USA; 4Orthopedic Surgery & Sports Medicine, Phoenix Children’s Hospital, Phoenix, AZ 85016, USA

**Keywords:** forward-trunk posture angles, oversride angles, foot strike patterns, mechanics

## Abstract

The current study aims to compare the mechanical propensities between healthy runners and runners with hamstring injuries. Retrospective case-control video analysis was used. A total of 35 (12 male and 23 female) videos of runners with hamstring injuries were compared with videos of sex-, age-, mass-, and height-matched healthy control runners. The main outcome variables were trunk posture angles, overstride angles, and foot strike patterns. An independent *t*-test and chi-squared tests were employed to analyze the main outcome variables between the runners with hamstring injuries and the healthy control runners. The statistical significance of less than 0.05 (*p* < 0.05) was used. The runners with hamstring injuries had a 1.6° less forward-trunk posture angles compared with the healthy control runners (*p* = 0.043). Also, the runners with hamstring injuries demonstrated a 4.9° greater overstride angles compared with the healthy control runners (*p* = 0.001). Finally, the runners with hamstring injuries had a tendency of rearfoot strike, while the healthy control runners showed a forefoot strike pattern (*p* = 0.004). In conclusion, the runners with hamstring injuries demonstrated different running mechanical propensities compared with the healthy runners.

## 1. Introduction

Running can generate numerous health benefits, including an enhancement in aerobic fitness, metabolic function, and postural balance [1]. However, as with every sporting activity, running entails a risk of sustaining a musculoskeletal pathology. According to an investigation performed by Taunton et al., hamstring injuries are one of the most common running-related pathologies [2]. Hamstring musculatures consist of semitendinosus, semimembranosus, and biceps femoris [3], and are known to have a high likelihood of recurrence [4,5,6]. A previous review article considered inadequate warm-up, excessive fatigue, lack of flexibility, agonist/antagonist imbalance, age, lower-back pain, sacroiliac-joint dysfunction, and chronic hormone imbalance as possible contributing factors for recurrent hamstring injuries [7]. 

In order to understand the mechanisms of hamstring injuries, a compelling exchange of ideas has been documented [8,9,10,11]. Additionally, a few studies have been performed in order to determine the risk factors of hamstring injuries [7,12,13,14,15]. A study performed by Heiderscheit et al. indicated that runners’ hamstring-strain injuries often occur at the late-swing phase of the running cycle, when the hamstrings are in a maximally stretched condition [15]. In addition to this finding [15], several studies have examined the effect of the proximal segment, such as the trunk, on hamstring injuries. Muckle postulated a clinical link between lumbar abnormalities and hamstring strain [16]. Also, Cibulka et al. identified a relationship between sacroiliac-joint dysfunction and hamstring-muscle strain [17]. Furthermore, Hennessey et al. found a poorer lower-back posture in hamstring injured athletes compared with non-hamstring injured athletes [18]. These reports suggest that hamstring injuries may be influenced by proximal segments, including the hip and trunk. From an anatomical standpoint, it may be rationalized that excessive lumbar lordosis may induce an anterior pelvic tilt, which may further increase the tension and stress of the hamstring musculature [19].

Another potential mechanism of hamstring injury could be related to the distal segments, including foot strike. Rearfoot strikers were found to sustain more than twice the rate of repetitive stress-related injuries than forefoot strikers (69% in rearfoot strikers vs. 31% in forefoot strikers) [20]. It has also been described that longer stride lengths require greater ground reaction force attenuation capabilities [21,22]. Increases in stride length entail increased knee extensions, which also facilitates in the elongation of the hamstring musculature. One study that analyzed stride frequency and foot position concluded that avoiding the so-called overstride [23], and positioning the foot relatively vertically to the tibia at the end of the swing phase is beneficial [24]. In short, both proximal and distal mechanics may be related to hamstring-injury mechanisms. However, the effects of both the proximal and distal segments on running mechanics are understudied, especially the roles of the trunk, lower leg/shin, and foot, although their functional roles have been theorized in an article [9]. Therefore, the aim of the current study was to identify running mechanical propensity differences, including the trunk, lower leg/shin, and foot positions, in runners with hamstring injuries compared with healthy runners. Our hypothesis was that the mechanical propensities of runners with hamstring injuries would be different from that of healthy runners. Specifically, runners with hamstring injuries would demonstrate greater forward-trunk postures and overstride angles. Also, runners with hamstring injuries would manifest more rearfoot-strike patterns than healthy runners. 

## 2. Methods

### 2.1. Study Design

A retrospective case control design with video analysis was employed to attain the current study’s aim. The setting in which this study was conducted was an injury-prevention laboratory affiliated with a sports medicine clinic of a regional tertiary-level pediatric medical center. The Institutional Review Board at Boston Children’s Hospital gave the approval for the current project, prior to the start of the current study. 

### 2.2. Procedures

The medical records of hamstring-injured patients who visited Boston Children’s Hospital over the least five years were extensively reviewed. The following inclusion criteria were used: (1) diagnosis of hamstring strain, (2) the hamstring diagnosis was achieved by a combination of physical examination and magnetic resonance imaging (MRI), (3) the hamstring-injured patients were engaging with running activities at the time of the clinical visits, and (4) the patients’ running images were captured at the time of the medical examination. The exclusion criteria were as follows: (1) no hamstring-strain diagnosis, (2) the absence of either physical examination or MRI, (3) non-engagement in running activities at the time of the clinic visit, and (4) missing running images. In addition to the four exclusion criteria, a previous history of surgery in the lower extremity regions, including the lower back, hip, thigh, knee, lower leg/shin, ankle, and foot, were excluded.

### 2.3. Running Protocol

The gait/running analysis was performed at The Micheli Center for Sports Injury Prevention as a part of their service. A gait/running analysis is routinely performed in order to evaluate the mechanics of injured and healthy runners at The Micheli Center for Sports Injury Prevention. The following procedures were used for the gait/running analysis: Participants performed a warm-up prior to running on a treadmill (Noraxon, Scottsdale, AZ, USA). The warm-up consisted of general stretching and slow-pace running on the treadmill. The entire warm-up session usually took approximately 5 min. Generally, the quadriceps, hamstrings, lateral thigh musculature, and calves were routinely stretched using self-selected formats. Following the warm-up, each participant selected their treadmill speed based on their comfort level. Video recordings began after the participants reported an adequate warm-up status and felt comfortable running on treadmill, which was usually during the middle of the running test. The treadmill speeds varied, but all of the participants ran at a speed in the range of 4–5 mph (6.44–8.05 km/h). This particular speed was selected because it usually facilitates a running motion without any discomfort or pain on the lower extremities. All of the recordings were performed with a treadmill slope grade of 0 (horizontal to the ground). 

### 2.4. Instrumentations

Unobstructed images were recorded by a high-speed camera, with rates of 300 frames per second, on a level treadmill. The camera used for this study was a Casio Exlim 1 at 300 fps (Casio, llc, Tokyo, Japan). The total video lengths varied from 10 to 60 s. Sagittal plane images were taken from the side. The camera was located 2.5 m away from the center of the treadmill. 

### 2.5. Operational Definitions

The following definitions were used to define the trunk posture angles, overstride angles, and foot strike patterns.

Trunk posture angles: Angles generated between a vertical line drawn from the superior tip of the greater trochanter to the ceiling, and to the acromioclavicular joint, from a side view (sagittal plane), at the moment of initial contact in a step, were defined as trunk posture angles. The angle measurements from the anterior relative to the superior tip of the greater trochanter were considered positive angles. The angle measurements of the posterior relative to the greater trochanter were considered negative angles (Figure 1).Overstride angles: The angles generated between a vertical line drawn from the fibular head to the ground, and to the lateral malleolus of ankle from a side view (sagittal plane), at the moment of initial contact in a step, were defined as the overstride angles. The angle measurements of the anterior relative to the fibular head were considered positive angles. The angle measurements of the posterior relative to the fibular head were considered negative angles (Figure 2).Foot strike patterns: When the rear part of the foot made contact on the treadmill from a side view (sagittal plane) at the moment of initial contact in a step, it was defined as the rearfoot strike (RFS). When the fore part of the foot made contact on the treadmill from a side view (sagittal plane) at the moment of initial contact in a step, it was defined as forefoot strike (FFS). When the middle part of the foot made contact on the treadmill from a side view (sagittal plane) at the moment of initial contact in a step, it was defined as midfoot strike (MFS). 

### 2.6. Data Analysis

Matched healthy runners were selected based on the injured runners’ sex, age, and body mass index (BMI). The running images of the healthy runners were extracted from the gait/running analysis database. The running images (trunk posture angles, overstride angles, and footstrike patterns) of the healthy runners were collected in the same manner described in the procedures and running protocol sections. Only the right limbs of the runners with hamstring injuries and of the healthy control runners were analyzed, because the camera was only located on the right side of the runners, and images of left limbs were difficult to capture and interpret. Thus, the runners who had hamstring injuries on their left limbs were treated as missing data (exclusion criteria 4) and were excluded from the data analysis. Then, a blinded video review was performed. All of the information that potentially disclosed the runners’ physical status (injured vs. healthy) were removed in the video data analysis. Also, in order to minimize bias, the video images were randomly organized by another author (D.M.). The randomization was performed using the Microsoft Excel (2013, Microsoft, Redmond, WA, USA) function. The video images were analyzed using ImageJ software (U.S. National Institutes of Health, Bethesda, MD, USA), by an engineer who dedicated their time to assisting in this project (D.D.). The author who performed the statistical analysis (D.S.) was also blinded from both the video selection and the video analysis processes. Blinding was maintained until the completion of the data analysis. 

### 2.7. Statistical Analysis

The demographics of the patients, including their age, height, mass, and body mass index (BMI), were analyzed descriptively. The dependent variables that were expressed as continuous variables, including the trunk posture angles and overstride angles, were analyzed between hamstring-injured and healthy runners, using an independent *t*-test if parametric shapes were observed. When non-parametric patterns were observed, a Mann–Whitney U test was used. The dependent variables that were categorical in nature, such as the foot strike patterns, were analyzed by a chi-squared test and were expressed as percentages (%). A *p*-value of less than 0.05 (*p* < 0.05) was used as the critical statistical value. In addition to the critical statistical value of *p* < 0.05, a 95% of confidence interval and effect size were incorporated as the continuous variable. The following effect sizes were used based on the recommendations of Cohen [25]: small effect 0.2–0.3, medium effect 0.5, and large effect ≥ 0.8. A series of the analyses were performed using SPSS version 23. (SPSS, Inc, Chicago, IL, USA).

## 3. Results

A total of 70 video images were analyzed. The group of runners with hamstring injuries (n = 35), and the age-, height-, mass-, and BMI-matched healthy control group (n = 35) each consisted of 12 males and 23 females (Table 1). Because the Shapiro–Wilk test indicated a parametric shape of the data, an independent *t*-test was employed. The runners with hamstring injuries had 1.6° less forward trunk posture angles compared with the healthy control runners (*p* = 0.043; Table 2). The effect size was approximately medium (Cohen’s d = 0.49). The runners with hamstring injuries demonstrated a 4.9° greater overstride angle compared with the healthy control runners (*p* = 0.001; Table 2). A large effect size was noted (Cohen’s d = 0.98). Nearly 75% of the runners with hamstring injuries demonstrated a propensity for an RFS pattern, while this was show for approximately only 43% of the healthy control runners. On the other hand, the tendency for FFS strike was about 6% in the runners with hamstring injuries, whereas roughly 35% of the healthy runners demonstrated FFS. These differences were reflected as being statistically significant in a chi-square analysis (*p* = 0.004; Table 3). 

## 4. Discussion

The primary aim of the current study was to identify the mechanical propensity differences between healthy runners and runners with hamstring injuries. It was hypothesized that runners with hamstring injuries would demonstrate a greater forward trunk posture and overstride angles with RFS patterns compared with healthy runners. This hypothesis was partially supported as follows: the runners with hamstring injuries showed greater overstride angles than the healthy runners (Table 2). Also, the RFS patterns were more common in the runners with hamstring injuries than in the healthy runners (Table 3). On the other hand, our results did not indicate greater forward trunk posture angles in the runners with a hamstring injury compared with healthy runners (Table 2). Instead, the runners with hamstring injuries demonstrated a significantly less forward trunk posture compared with healthy runners (Table 2). 

Several past studies have synthesized the mechanical contribution of hamstring musculature on injury during running. One of the studies found that hamstring injuries occur because of the large amount of time that the muscle group spends in a maximally stretched condition [15]. Another study reported that maximum tension is placed on the hamstring tendons when the hip is flexed and the knee is extended [26]. Opar et al. summarized that repetitive eccentric loading in maximal hip flexion and knee extension during running may potentially lead to microscopic muscle damages, and when the damages exceed the mechanical limits, repetitive microscopic damage likely turns into an injury [6]. The findings of this study partially support the mechanisms of hamstring injuries reported by past studies [6,15,26]. While the foot lands on ground with overstride mechanics, the hamstring musculature may be stretched eccentrically, which may facilitate in increasing the tension on the hamstring muscle bundles. According to Hunter and Faulkner, the magnitude of the muscle fibers lengthened by a force such as stretching is the most accurate indicator of eccentric muscle injury [27]. Moreover, it was estimated that hamstrings need to counterbalance approximately 10 times of the body weight in order to control the locomotion during running [11]. Unfortunately, the current study did not have measurements in the amount of muscular stretching, potential damage on muscle fibers, and the force generated in running. Thus, our results cannot be generalized to those components [11,27], and is only limited to postulate the potential mechanical propensity differences between runners with hamstring injuries and healthy runners. 

Additionally, our results showed that runners with hamstring injuries had a propensity for RFS in running compared with healthy runners (Table 3). According to a study performed by Kuhman et al., a greater loading rate of the ground reaction impact peak and an increased ankle dorsiflexion moment were found in the runners with habitual RFS [28]. In a recent meta-analysis, this high loading of the ground reaction force was identified as a risk factor of running, related musculoskeletal injuries [29]. Furthermore, another study conducted by Daoud et al. reported that musculoskeletal injuries were more commonly found among cross-country runners with RFS than cross-country runners with MFS or FFS [20]. To synthesize the reported evidence and current study results, it can be rationalized that the higher ground reaction force created by the RFS patterns and greater overstride angle are transmitted to the hamstring muscular unit during its most susceptible time, when it is at greatest stretch at footstrike, which may be related to the hamstring injury mechanism. With regards to hamstring tissue stretch capability and injury, van Mechelen et al. examined the effect of a 16-week stretching program using a randomized controlled trial [30]. This study did not find statistical differences in the running related injuries between intervention and control groups [30]. Therefore, more investigations are necessary in order to find an association with running related injuries and various running parameters, including foot strike types, hamstring flexibility, and mechanical propensities.

It was unexpected that runners with the hamstring injuries displayed a decreased forward trunk posture angle compared with healthy runners (Table 2). Our hypothesis was developed because multiple authors in the past have reported a clinical and mechanical link between the proximal segments, such as the trunk and hip, and hamstring injuries [16,17,18,31]. Specifically, Hennessey et al. found that increased lumbar lordosis is associated with hamstring strain injury [18]. Furthermore, a cross-sectional laboratory study conducted by Kulas et al. suggested that greater trunk flexion angles increase hamstring activation [31].The same author also reported that a greater trunk extension posture facilitates more quadricep-dominant dynamic movements [32]. These studies substantially influenced us in developing our hypothesis, and we postulated that greater forward trunk posture angles would be observed in runners with hamstring injuries, because a forward trunk posture emphasizes stretching the hamstrings further, based on the proximal anatomical origin at the ischial tuberosity. However, the findings from our study contradicted this notion [18,31]. One of the potential reasons between the past reports and the current findings may stem from the running speed used. Hamstring injuries often occur at a high speed, such as when sprinting [33,34,35]. However, the speed of our gait/running analysis was set at 4–5 mph (6.44–8.05 km/h). The fairly slow speed used in the gait/running protocol might not be enough to induce the positioning of proximal segments that is often reported as a mechanism of hamstring injuries [33,35,36]. This might potentially explain the underlying mechanism of the decreased forward trunk posture angles in runners with hamstring injuries observed in the current study. 

### Limitations

The limitations of this study need to be explained. First, the current study was a retrospective analysis of cross-sectionally obtained data. Ideally, a prospective study design needed to be used. For instance, taking running images in a pre-season or pre-participation physical, and following runners’ musculoskeletal pathology status, including hamstring injuries, over time would provide a strong internal validity between the mechanical variables and hamstring injuries. Furthermore, when the data collection was performed, the runners already had some type of pain and discomfort in their hamstrings. Thus, it is unclear that whether the observed running propensity in runners with hamstring injuries was the cause or the result of the injury. In order to address this issue, more scientifically rigorous study designs are necessary in future studies. Second, the data were not analyzed based on each sex (males and females separately) in the current study. One recent study identified sex differences in running mechanics and coordination variability [37]. Thus, analyzing the current data based on each sex, or comparing the differences by sex would be an interesting approach. However, the number of male runners with hamstring injuries was relatively small. Thus, we did not pursue this opportunity. Third, there is variability in runners with hamstring injuries. Most runners with hamstring injuries had pain or discomfort for a while. However, the duration of the pain or discomfort; status of the prior rehabilitation; and other types of training, such as resistance training, varied in the study population. The last limitation was related to data collection. The current study was conducted on a treadmill. Several studies have reported that running mechanics are different between treadmill and on-the-ground running [38,39]. Additionally, the significant findings of the current study were all from the sagittal plane, including forward trunk flexion angles, overstride angles, and foot strike foot patterns. Therefore, the effects of coronal/frontal and horizontal planes were not included in the current study. Future studies are warranted in order to incorporate all three planes (sagittal, coronal/frontal, and horizontal planes) so as to analyze a comprehensive running mechanical propensity.

## 5. Conclusion

The purpose of the current study was to compare the running mechanical propensities between runners with hamstring injuries and healthy runners. The results of the study indicated that he runners with hamstring injuries tend to have a propensity for overstride angles and RFS patterns compared with healthy runners. The decreased forward trunk flexion posture observed in runners with hamstring injuries may be related to the fairly slow running speed used in the current study protocol. Despite a few limitations, the findings of this study may be useful for understanding the mechanisms of hamstring injuries. Future studies need to investigate running and musculoskeletal injuries, such as hamstring strain, with a superior study design, including various parameters based on specific populations (for example, males and females separately). 

## Figures and Tables

**Figure 1 sports-07-00210-f001:**
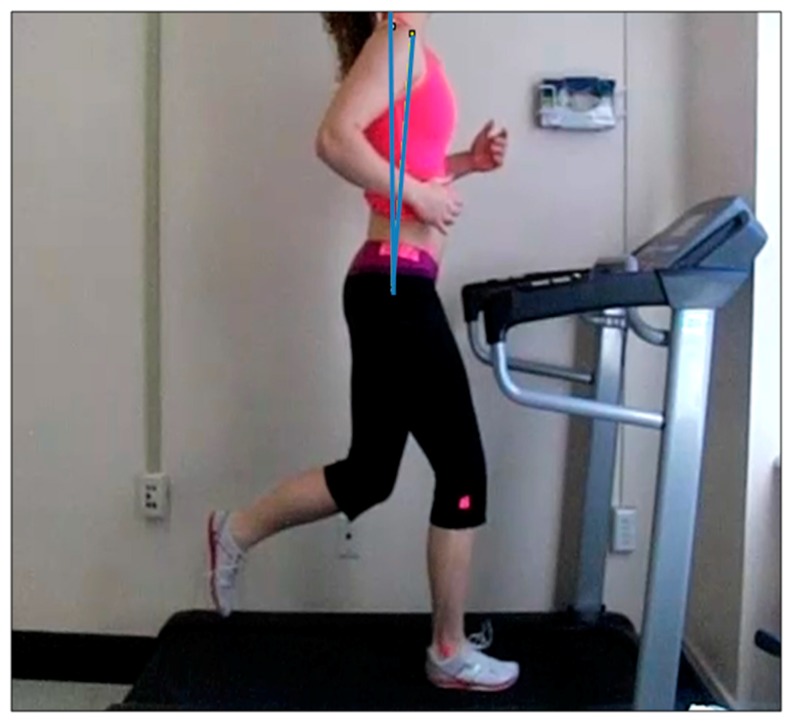
Image of the trunk posture angles. The angles generated between the vertical line drawn from the superior tip of the greater trochanter to the ceiling, and to the acromioclavicular joint, from a side view (sagittal plane), at the moment of initial contact in a step, are defined as the trunk posture angles. The angle measurements of the anterior relative to the superior tip of the greater trochanter are considered positive angles. The angle measurements of the posterior relative to the greater trochanter are considered negative angles.

**Figure 2 sports-07-00210-f002:**
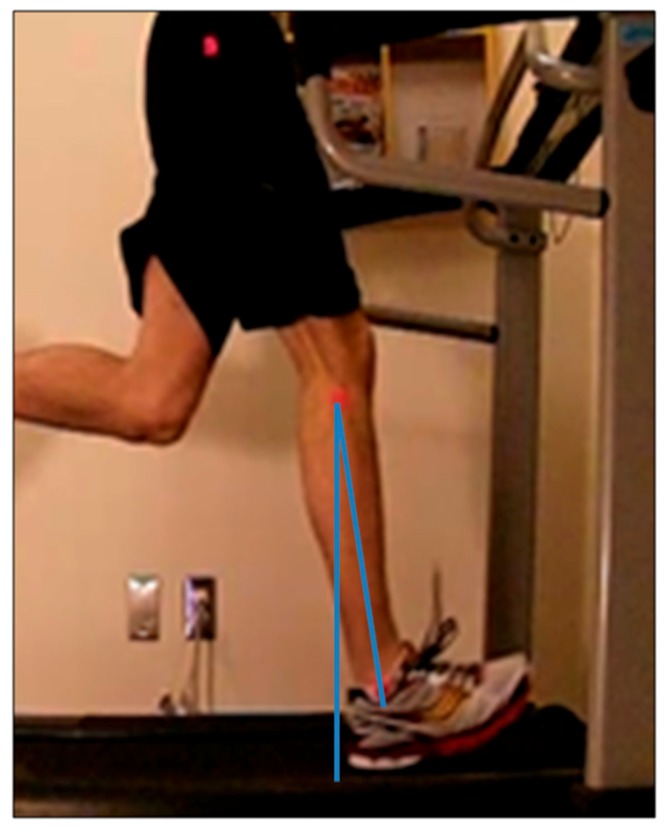
Image of overstide angles. The angles generated between a vertical line drawn from the fibular head to the ground, and to the lateral malleolus of the ankle, from a side view (sagittal plane), at the moment of initial contact in a step, were defined as the overstride angles. The angle measurements of the anterior relative to the fibular head are considered positive angles. The angle measurements of the posterior relative to the fibular head are considered negative angles.

**Table 1 sports-07-00210-t001:** Physical characteristics of hamstring-injured runners and healthy runners.

Physical Characteristics	Hamstring Injured Runners (12 Males and 23 Females)	Healthy Runners (12 Males and 23 Females)	*p*-Value (<0.05)
Age (year)	29.0 ± 12.4 (24.7; 33.3)	29.1 ± 12.5 (24.9; 33.4)	0.966
Height (cm)	171.2 ± 9.6 (167.8; 174.5)	167.1 ± 9.0 (164.0; 170.2)	0.072
Weight (kg)	64.7 ± 10.7 (60.9; 68.4)	60.9 ± 10.7 (57.2; 64.6)	0.149
BMI	22.0 ± 2.2 (21.2; 22.7)	21.7 ± 2.9 (20.7; 22.7)	0.687

Values are mean ± standard deviation. The 95% confidence interval (CI) values are expressed within the brackets. All of the hamstring injuries were on the lateral side (23 biceps femoris) or medial side (12 either semitendinosus or semimembranosus).

**Table 2 sports-07-00210-t002:** Comparison of trunk posture angles and overstride angles between hamstring injured runners and healthy runners.

Angles	Hamstring Injured Runners (12 Males and 23 Females)	Healthy Runners (12 Males and 23 Females)	*p*-Value (<0.05)
Trunk posture angle (degrees)	2.9 ± 2.5 (2.1; 3.8)	4.5 ± 3.9 (3.2; 5.9)	0.043 *
Overstride angle (degrees)	2.9 ± 3.9 (1.6; 4.2)	−2.0 ± 5.9 (−4.0; 0.1)	0.001 *

Values are mean ± standard deviation. The 95% CI values are expressed within the brackets. * *p* < 0.05. Effect size: trunk posture angles: Cohen’s d = 0.49 (approximately medium effects of 0.50); overtride angles: Cohen’s d = 0.98 (greater than large effects of ≥ 0.80).

**Table 3 sports-07-00210-t003:** Comparison of foot strike patterns between healthy runners and hamstring injured runners.

Foot Strike Types	Hamstring Injured Runners (12 Males and 23 Females)	Healthy Runners (12 Males and 23 Females)	Total
Rearfoot strike	26 (74.3%)	15 (42.9%)	41 (58.6%)
Midfoot strike	7 (20.0%)	7 (20.0%)	14 (20.0%)
Forefoot strike	2 (5.7%)	13 (35.1%)	15 (21.4%)
Total	35 (100.0%)	35 (100.0%)	70 (100.0%)

The values are frequencies. The proportions of each foot strike type are expressed within the brackets. Chi-squared analysis: *p* = 0.004.

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
