# Peer review of "Running Propensities of Athletes with Hamstring Injuries"

_sports, 2019, doi:10.3390/sports7090210_

Round 1

Reviewer 1 Report

There are studies on running mechanics and hamstring strains that are not included in this study. Authors need to consider that the current debate on the exact phase where hamstring injury occurs, see:

Journal of Sport and Health Science 6 (2017) 130–132

Journal of Sport and Health Science 6 (2017) 133–136

Journal of Sport and Health Science 6 (2017) 137–138

Lines 71-73: More information regarding the patients should be provided. How long were the athletes injured prior to testing? An athlete injured 5 years ago may not have the same clinical picture than another athlete who got injured 1.5 year ago. In addition, the intensity of injury should be reported. Finally, did all players follow rehabilitation programs?

Lines 167-169: How do the results of this study confirm previous studies on injury mechanics? Neither stretch was measured, nor muscle damage were monitored.

Lines 173-176: Stress fracture varies significantly from hamstring injury. Furthermore, it is not clear whether athletes had the altered RFS pattern prior to injury or they display this pattern after injury.

Lines 185-186: Yes, but there is also evidence that hamstring flexibility is not related to hamstring strain. Please see the review by Opar (No 6 in reference list).

Line 203: Additional limitation males and females

It would be more interesting if you could present us which of the muscles specifically were injured(biceps, semitendinosus, semimebranosus,maybe even popliteus ?(although it doesn't belong to hamstrings) since you have an MRI for each patient

Reviewer 2 Report

This paper aimed to identify running mechanical, trunk and overstride angles and foot strike pattern, differences between runners with hamstring injuries and healthy runners. 

My main issue is with measures chosen and the discussion. I believe a much more in depth explanation of why the mechanical variables were assessed and how important the observed effects are to practice etc. Also, I am not entirely convinced of the impact to practitioners.

Additionally, the entire manuscript needs to be edited for grammar.

Please see my specific comments below.

Abstract

Line 20: Consider removing the words "As the Results,".

Introduction

Line 44: "Hennessey" No citation is included with this. It is included later in the manuscript. Include a citation and revise the reference numbers.

Line 47-49: Please support this with literature.

Line 51: "significantly higher" How much higher? can you include a %.

What variables are of primary interest? This should be covered in the introduction.

Methods

Line 91-92: Word missing.

Line 92-93: Could you include 4-5 mph as km/hr as well. Why was this speed used? Please justify it.

Line 95-96: "300 frames per seconds (fps) to 300 fps" This line does not make sense?

Line 100-114: Please revise for grammar.

Line 100-114: Why only these three variables?

Line 116-117: Consider reversing this sentence to improve clarity.

Line 119-120: Please revise for grammar.

Line 120-122: Can you give an example of this?

Line 123-127: Why are the abbreviations DM, DD and DS included? They don't appear to be necessary.

Line 135: Please revise for grammar.

Line 137-138: Could you rewrite these effect sizes to make them clearer.

Results

Line 148: The greatest number of of runners had a rear foot strike in the healthy group as well.

Line 180: Typo "muscularunit" 

Limitations

Line 206: "retrospective" Should this be cross sectional?

Tables: Numbers in brackets are 95% CI? Please state this in the caption or underneath the tables.

Figures 1 & 2: Caption goes underneath.

Round 2

Reviewer 2 Report

I am spell check of the entire manuscript as there are countless grammatical errors throughout.

Author Response

Thank you. We have revised the whole manuscript for English corrections.